# Does Impaired Plantar Cutaneous Vibration Perception Contribute to Axial Motor Symptoms in Parkinson’s Disease? Effects of Medication and Subthalamic Nucleus Deep Brain Stimulation

**DOI:** 10.3390/brainsci13121681

**Published:** 2023-12-06

**Authors:** Tobias Heß, Peter Themann, Christian Oehlwein, Thomas L. Milani

**Affiliations:** 1Department of Human Locomotion, Chemnitz University of Technology, 09126 Chemnitz, Germany; 2Department of Neurology and Parkinson, Clinic at Tharandter Forest, 09633 Halsbruecke, Germany; 3Neurological Outpatient Clinic for Parkinson Disease and Deep Brain Stimulation, 07551 Gera, Germany

**Keywords:** Parkinson’s disease, axial motor symptoms, postural instability and gait difficulties, functional limits of stability, non-motor sensory symptoms, sensorimotor integration, somatosensory system, plantar cutaneous vibration perception thresholds, deep brain stimulation, subthalamic nucleus

## Abstract

Objective: To investigate whether impaired plantar cutaneous vibration perception contributes to axial motor symptoms in Parkinson’s disease (PD) and whether anti-parkinsonian medication and subthalamic nucleus deep brain stimulation (STN-DBS) show different effects. Methods: Three groups were evaluated: PD patients in the medication “on” state (PD-MED), PD patients in the medication “on” state and additionally “on” STN-DBS (PD-MED–DBS), as well as healthy subjects (HS) as reference. Motor performance was analyzed using a pressure distribution platform. Plantar cutaneous vibration perception thresholds (VPT) were investigated using a customized vibration exciter at 30 Hz. Results: Motor performance of PD-MED and PD-MED–DBS was characterized by greater postural sway, smaller limits of stability ranges, and slower gait due to shorter strides, fewer steps per minute, and broader stride widths compared to HS. Comparing patient groups, PD-MED–DBS showed better overall motor performance than PD-MED, particularly for the functional limits of stability and gait. VPTs were significantly higher for PD-MED compared to those of HS, which suggests impaired plantar cutaneous vibration perception in PD. However, PD-MED–DBS showed less impaired cutaneous vibration perception than PD-MED. Conclusions: PD patients suffer from poor motor performance compared to healthy subjects. Anti-parkinsonian medication in tandem with STN-DBS seems to be superior for normalizing axial motor symptoms compared to medication alone. Plantar cutaneous vibration perception is impaired in PD patients, whereas anti-parkinsonian medication together with STN-DBS is superior for normalizing tactile cutaneous perception compared to medication alone. Consequently, based on our results and the findings of the literature, impaired plantar cutaneous vibration perception might contribute to axial motor symptoms in PD.

## 1. Introduction

Parkinson’s disease (PD) is a prevalent age-related and progressive neurodegenerative disorder and its primary cause is associated with the decline of dopaminergic neurons within the substantia nigra pars compacta [1,2,3,4,5]. The dopamine deficiency leads to a hypo-dopaminergic state in the basal ganglia, causing an imbalance in excitatory and inhibitory communication within the nigro-striatal and thalami-cortical circuits. As a consequence, it affects the network of brain structures responsible for planning, selecting, adapting, and executing motor functions [5,6,7,8]. After losing a significant portion of about 80% of dopamine-producing cells, clinical motor symptoms, including tremor, rigidity, bradykinesia, postural instability, and gait difficulties become noticeable [9,10,11]. Although treatments such as dopamine replacement therapy or deep brain stimulation (DBS) of the subthalamic nucleus (STN) can relieve axial motor symptoms, their effects on postural and gait control are still controversial [8,12,13,14,15].

While those motor symptoms can be very prominent and cause a significant loss in patient quality of life, PD can also be accompanied by numerous non-motor sensory symptoms [16,17,18,19]. Among those non-motor sensory symptoms, defective functionality of sensory systems such as visual [20,21,22,23], vestibular [8,24,25,26,27,28], and somatosensory systems [29,30,31,32,33,34,35,36,37,38] has been reported in PD patients. Since those sensory systems provide valuable information for controlling movements, impaired sensory system functionality might therefore contribute to PD patients’ motor symptoms. This might especially be true as those sensory systems gather information about the surrounding environment, the position of the body in space, and the relative position of each joint with respect to other parts of the body, as well as information about the condition of the ground on which the person is standing or walking [23,24,39,40]. Given that only the soles of the feet are in direct contact with the ground during standing or walking, particularly afferent information from plantar cutaneous mechanoreceptors appears to be crucial for motor control. Plantar cutaneous mechanoreceptors gather information about the pressure distribution and loading shifts underneath the foot during movements, and therefore are involved in adapting muscle contraction tone and contraction patterns [41,42,43,44,45,46,47,48,49,50,51,52]. The sensorimotor integration of plantar mechanoreceptors has already been investigated in several studies with individuals without neurological diseases. Those studies have shown that decreased plantar cutaneous sensation achieved by anesthesia of the foot sole led to impaired control of static and dynamic balance abilities, as well as gait performance [42,46,49,50,51,52,53,54,55,56]. Conversely, several studies have demonstrated that sensory stimulation of the foot sole using various types of shoe insoles improved balance and gait performance in PD patients [41,43,44,45]. Accordingly, those findings suggest that PD patients might suffer from impaired plantar tactile cutaneous perception, while mechanical stimulation facilitates the sensory signal and therefore enhances sensorimotor integration.

This makes sense from a pathophysiological point of view because the mechanisms of sensory symptoms in PD include findings of widespread deposits of α-synuclein, which is a fundamental pathological protein and a major component of Lewy bodies in PD [16,57,58,59,60,61,62]. Impaired plantar tactile cutaneous perception in PD patients is therefore quite conceivable, since it has been shown that α-synuclein affects numerous sensory-related structures of the central nervous system, as well as the peripheral nervous system, including the skin [57,63,64,65,66,67,68]. This is supported by the study from Nolano et al., who investigated skin biopsies and found peripheral denervation in PD patients. More specifically, the authors reported demyelination of epidermal nerve fibers of the glabrous and the hairy skin, as well as a significantly reduced number of mechanoreceptors in PD patients compared to those of healthy subjects. In particular, Meissner corpuscles, which detect mechanical vibration stimuli, may be affected, whereas the loss of those mechanoreceptors correlated with the disease severity of the patients. The authors concluded that peripheral deafferentation could play a major role in the pathogenesis of sensory dysfunction in PD [64]. Besides results showing impairments of the peripheral nervous system, there is also evidence for the defective central integration and processing of afferent information at a cerebral level in PD patients. Although the basal ganglia are considered well-established, primarily motor-related structures, they might also function as an active “sensory analyzer” for higher-level central somatosensory processing [69,70,71,72,73]. This is plausible, since the basal ganglia have projections into the thalamus and the cortex and receive input not only from motor areas, but also from cortical somatosensory areas, including the primary and the secondary somatosensory cortices [69,70,73,74]. Using 3D positron emission tomography, Boecker et al. showed that basal ganglia dysfunction in PD is characterized by abnormal sensory processing even for tasks devoid of any motor component [73]. Consequently, impaired tactile cutaneous perception in PD might be driven by both denervation of peripheral epidermal nerve fibers and mechanoreceptors and the defective central integration and processing of afferent information due to the diseased basal ganglia network [38,64,73,75,76,77,78,79,80,81,82,83,84,85,86].

Despite this clinical and experimental evidence of cutaneous denervation and impaired sensory processing in PD, investigations about tactile cutaneous perception are rare, especially for the foot sole, and the few existing studies show rather discordant results. While some studies found impaired plantar tactile cutaneous perception in PD patients [64,87,88], others failed to find PD affecting cutaneous thresholds for mechanical stimuli of the foot [48,89]. Lack of studies and contradicting results are also prominent for investigations about the effect of various therapies, such as anti-parkinsonian medication and STN-DBS for patients’ tactile cutaneous perception of the foot. According to the literature, anti-parkinsonian medication seems to have generally minor to no effects [48,64,87,89]. On the other hand, STN-DBS generally seems to be more promising for treating tactile cutaneous perception in PD [32,33,34,36,37,81,90,91,92]. However, existing studies have only focused on investigations of the upper body, including the torso, arms, hands, or fingers. To the best of our knowledge, no other studies have explored the effect of STN-DBS on tactile cutaneous perception in PD patients’ feet.

Specifically investigating plantar tactile cutaneous perception in PD patients could be beneficial for several reasons. It has been reported that sensory symptoms may occur prior to the presence of motor symptoms, so they could be used for early diagnosis of PD [16,93,94,95]. Moreover, this may help to distinguish between PD and various other neurodegenerative diseases, such as multiple system atrophy, which has been demonstrated with a high sensitivity and specificity of approx. 80% [87]. Consequently, testing plantar cutaneous sensation could serve as a potential surrogate marker. It may also add to our understanding of how PD affects the peripheral nervous system, as well as offer insight into sensorimotor integration and potentially help to better understand the cause of axial motor symptoms in PD. Thinking ahead, plantar tactile cutaneous perception measurements could help to develop and optimize low-budget therapy devices, such as textured insoles, to stimulate plantar cutaneous mechanoreceptors and consequently enhance motor performance in PD patients [41,43,44,45]. As this is the first study investigating the effects of STN-DBS on plantar tactile cutaneous perception in PD patients, it may provide valuable information for better understanding the neurophysiological processing of afferent sensory input. Furthermore, it could assist in counseling patients regarding suitable treatments and support clinicians in customizing and optimizing therapy strategies. 

Therefore, we pursued two objectives with this study. 

First, we investigated and characterized axial motor symptoms in PD patients, such as postural instability and gait difficulties. We examined whether PD patients treated with anti-parkinsonian medication and STN-DBS have superior motor control compared to patients treated with medication alone. We also tested healthy elderly subjects as a reference. We hypothesized that patients’ motor performance is worse compared to that of healthy subjects, and that the combination of medication and STN-DBS offers greater advantages in normalizing patients’ abnormal motor control compared to treatment with medication alone. 

Second, we investigated somatosensory functionality by analyzing plantar cutaneous vibration perception thresholds (VPT) within the same study groups. Based on the pathophysiological mechanisms of PD and previous study findings, we hypothesized that plantar cutaneous vibration perception of patients would be impaired compared to that of healthy subjects. Furthermore, we assumed that medication in combination with STN-DBS would show superior effects on normalizing patients’ impaired plantar cutaneous vibration perception compared to treatment with medication alone.

## 2. Methods

### 2.1. Subjects

Three distinct study groups underwent evaluation: Patients with Parkinson’s disease (PD-MED), patients with Parkinson’s disease who previously underwent deep brain stimulation surgery (PD-MED–DBS), as well as healthy subjects. Both patient groups included subjects diagnosed with idiopathic Parkinson’s disease according to the diagnostic criteria of the Movement Disorders Society. They had to be at least 50 years old and have a disease severity between 2 and 3 on the Hoehn and Yahr scale [96]. All recruited PD patients primarily showed signs of postural instability and gait difficulty subtype of the disease. Subjects assigned to group PD-MED–DBS were required to have undergone bilateral high-frequency (≥130 Hz) deep brain stimulation surgery of the STN at least one year prior to guarantee optimized DBS settings and maximum efficiency [15,97]. The cut-off for the duration of DBS since surgery was defined as 5 years since studies have demonstrated a gradual decrease in stimulation efficacy over time [98,99,100]. All DBS patients included in this study showed a positive response to the surgery. Subjects of both patient groups underwent the examinations in the medication “on” state while under the influence of regular anti-parkinsonian medication, including levodopa. Patients in the PD-MED–DBS study group were additionally in the “on” stimulation state. The exclusion criteria for both patient groups comprised secondary pathologies affecting the motor and somatosensory systems, such as atypical parkinsonism, severe camptocormia, severe tremor, diabetes mellitus with polyneuropathy and normal pressure hydrocephalus. All patients had to be able to stand and walk without assistance. Patients with cognitive deficits (mini-mental state examination (MMSE) < 24/30), psychiatric issues, or severe depression were not considered for inclusion. All PD patients were recruited and underwent evaluations as part of the patient consultation at the Neurological Outpatient Clinic for Parkinson’s Disease and Deep Brain Stimulation in Gera, Germany, led by Christian Oehlwein. Relevant clinical data from the most recent neurological examination were provided for both patient groups (Table 1). The control group, which comprised healthy elderly subjects, was examined in the laboratory of the Department of Human Locomotion (Chemnitz University of Technology, Germany). The healthy subjects had no injuries or diseases, and took no medication that could have affected cognition, postural performance, or cutaneous perception.

### 2.2. Equipment and Testing Procedures

Before data acquisition, all subjects were briefed about the purpose of this study and provided written informed consent. All procedures were conducted according to the recommendations of the Declaration of Helsinki and were approved by the ethics committee of the medical faculty of the University Leipzig (IRB number: 023/14-ff, 2 April 2014).

#### 2.2.1. Motor Performance

Motor performance was quantified using a pressure distribution platform (Zebris FDM 1.5; Isny, Germany, sampling frequency 100 Hz). For subjects’ safety, the platform was integrated into a walkway so that it was level with the ground. All tests were carried out barefoot. Plantar skin temperature was monitored with an infrared thermometer (UNI-T UT302C, Augsburg, Germany), since alterations may affect tactile perception and, consequently, motor performance [46,56,101,102]. 

##### Quiescent Bipedal Stance

Subjects were required to maintain a still bipedal stance with an upright posture, keeping the knees straightened but not locked, allowing both arms hanging down loosely, and directing their gaze straight ahead (Figure 1A). Stance width could be chosen individually and was controlled measuring the heel-to-heel distance. Three consecutive trials lasting 20 s each were collected for each subject. 

##### Functional Limits of Stability

To detect subjects’ limits of stability, they were instructed to stand as still as possible for approx. 10 s to define the neutral COP location. Thereafter, subjects had to lean forward three times and backward three times as far as they could, using the ankle joints as the pivot point. They were neither allowed lifting of their heels or toes from the platform, nor were they allowed bending of their back or flexing their knee and hip joints (Figure 1B). When the subjects reached their maximum leaning posture, they were instructed to hold it for approx. 5 to 10 s [20,103,104,105,106]. The examiner stood nearby and carefully monitored the procedure and assisted in case of balance loss. If this happened, the trial was repeated. 

##### Gait

Subjects were instructed to walk over the platform at least 8 times to generate sufficient data. Gait speed had to be quick but comfortable and safe, and it could be chosen individually (Figure 1C).

#### 2.2.2. Plantar Vibration Perception

A customized vibration exciter (Tira Vib, TV51075, Schalkau, Germany) was used to quantify plantar cutaneous VPTs [47,107,108]. A vibrating brass probe (diameter 7.8 mm), extending through an aperture in an aluminum foot rest, applied sinusoidal vibration stimuli to the plantar foot (Figure 2). The vibration frequency was set at 30 Hz, targeting primarily fast adapting (FA I) cutaneous afferents (Meissner corpuscles) [109,110], which are known to be impaired in PD [64]. Vibration frequency and amplitude were verified using an accelerometer (NXP Semiconductor, MMA2241KEG, Eindhoven, Netherlands) attached to the probe. Given that alterations in skin temperature can influence tactile perception, a thermal element was affixed to the footrest, set at 20 °C to ensure comparability [46,102]. Before the examination, each subject underwent an acclimatization period lasting approx. 10 min. This involved placing their bare feet on an aluminum plate while comfortably seated in a chair, with the hip and knee joints bent at approx. 90°. Subsequently, the first metatarsal head of the left or right foot was positioned precisely perpendicular to the probe in a random order. Since alterations of the force the probe exerts against the skin affect tactile perception, the contact force was monitored throughout the measurement using a sensor (Hottinger, U9B, Darmstadt, Germany) [111,112]. A customized algorithm (written in Labview 2015, National Instruments, Austin, TX, USA), modified after Mildren et al. was implemented to automatically detect the VPT [113,114,115,116]. The algorithm started with a supra-threshold sinusoidal vibration burst (2 s duration), and patients indicated having perceived the vibration by pushing a handheld trigger. Consequently, the vibration amplitude was gradually reduced by 50% until the subject no longer perceived the burst. Then, the average amplitude from the last perceived and the last unperceived burst was tested. The algorithm ended four bursts after the first undetected stimulus. To prevent bias from anticipation and habituation, the algorithm randomly varied the pause time between two consecutive vibration stimuli (ranging between 2 and 7 s). The recorded VPT was defined as the mean of the smallest perceived and largest unperceived vibration amplitude in µm. Each subject underwent one test trial and three main trials for both feet. To eliminate environmental noises, subjects were required to wear noise-canceling headphones (QuietComfort 25, Bose GmbH, Friedrichsdorf, Germany). Plantar temperatures of both feet were monitored employing the same infrared thermometer used for the motor performance tests. 

### 2.3. Data Processing

#### 2.3.1. Motor Performance

All data underwent processing using a script written in MATLAB R2022a (Math-WorksTM, Natick, MA, USA).

##### Quiescent Bipedal Stance

Raw COP displacement data were processed using a recursive zero phase shift filter with a cut-off frequency of 35 Hz. Subsequently, the COP ranges, the COP maximum velocities (for both parameters in the anterior–posterior (AP) and medio–lateral (ML) directions), the COP 95% confidence area, and the COP maximum velocity total were calculated. For each COP parameter, the mean over the three collected trials for each subject was calculated and used for further statistical analysis.

##### Functional Limits of Stability

Raw COP displacement data were processed using a recursive zero phase shift filter with a cut-off frequency of 35 Hz. The COP signal was visually inspected, and the neutral COP location and the three anterior and posterior maximal COP displacements were identified manually. The total COP range was calculated by adding together the anterior and posterior COP displacements for each trial and subject. The neutral COP location, the maximal COP displacement, and the full range were normalized to the foot length of each subject [104,105,117]. Additionally, the motion times until maximal COP displacements were reached and the corresponding mean velocities for each trial were calculated (Figure 3). The mean over the three collected trials for each subject was calculated and used for further statistical analysis. 

##### Gait

Spatial–temporal gait parameters, such as stride length, stride width, stride time, gait velocity, cadence, double support time, and COP length during the stance phase of gait were exported from Software Zebris winFDM (version 0.1.11, Isny, Germany). The mean over all collected trials for each subject was calculated and used for further statistical analysis.

#### 2.3.2. Plantar Cutaneous Vibration Perception

All VPTs were transformed with the natural logarithm to achieve normality and correct for the naturally skewed distribution [108,118]. The mean over the last three collected VPTs was calculated for further statistical analysis.

### 2.4. Statistical Analysis

Statistical analysis was performed using SPSS Statistics (IBM, version 29.0.0.0, Ehningen, Germany). The Shapiro–Wilk test (α = 0.05) was used to check for data distribution. 

Differences between study groups were investigated using one-way analysis of variance for normally distributed data and the Kruskal–Wallis test for non-normally distributed data. To account for the study groups and testing conditions, the level of significance was Bonferroni-corrected individually. Effect sizes, r, were calculated as well. For the examination of the intra-group variability, an overall coefficient of variation for each test was computed. 

For intra-group comparisons involving sex, body side, disease-dominant side, and AP vs. ML directions, the *t*-test for paired or independent samples was used for normally distributed data. In the case of non-normally distributed data, the Wilcoxon test was used.

## 3. Results

### 3.1. Demographic and Clinical Data

As shown in Table 1, there were considerably more male than female subjects in all study groups. On average, patient group PD-MED–DBS comprised younger subjects compared to both other study groups, with statistically significant differences compared to HS. Self-rated balance confidence and gait confidence were lower for both patient groups compared to the healthy subject group, HS. No differences were found for clinical data MMSE, UPDRS III, UPDRS total, or Hoehn and Yahr ratings between PD-MED and PD-MED–DBS. On average, PD-MED–DBS suffered from Parkinson’s disease for more than twice as long as PD-MED. The dominant disease side was evenly distributed among patients in group PD-MED. However, in group PD-MED–DBS, there were more patients with disease dominance on the right side of the body. In PD-MED, the time interval between the most recent clinical examination and the motor and vibration tests was longer and showed higher variability, in contrast to PD-MED–DBS. The mean duration of DBS since surgery was 33.1 ± 25.7 months, and patients’ self-rated satisfaction with DBS at the time of the tests was 80.1 ± 20.6%.

### 3.2. Motor Performance

#### 3.2.1. Quiescent Bipedal Stance

Inter-Group Comparisons

Self-selected stance width did not differ between groups: PD-MED 13.1 cm ± 2.7 cm; PD-MED–DBS 13.7 cm ± 3.0 cm; HS 12.7 cm ± 2.0 cm. Statistically significant differences between study groups were only found for spatial COP parameters (Figure 4A–C), but not for COP velocities (Figure 4D–F). In more detail, PD-MED and PD-MED–DBS showed significantly higher COP areas with higher COP ranges in the AP and ML directions compared to HS. COP velocities only showed a trend towards higher values for PD-MED and PD-MED–DBS compared to HS. No statistically significant differences were found between PD-MED and PD-MED–DBS. The overall coefficient of variation showed higher variability for PD-MED (0.41 ± 0.12) and PD-MED–DBS (0.40 ± 0.11) compared to HS (0.34 ± 0.10).

Intra-Group Comparisons

Within study groups, no significant differences were found between male and female subjects or between patients with left and right disease-dominant body sides. However, statistically significant intra-group differences (*p* < 0.004) were found for each group between the AP and ML directions for COP ranges and COP max velocities. Consequently, all groups showed higher COP values in the AP direction compared to the ML direction. 

#### 3.2.2. Functional Limits of Stability

Inter-Group Comparisons

The neutral COP location showed comparability between all groups: PD-MED 39.4 ± 5.8%; PD-MED–DBS 38.0 ± 4.7%, and HS 41.6 ± 4.3% of the foot length measured from the heel. Various statistically significant inter-group differences were found for COP displacement. Both PD-MED and PD-MED–DBS showed smaller COP displacements in the posterior direction and reduced total COP ranges compared to HS. Only PD-MED showed smaller values for the COP displacement in the anterior direction, which were statistically significant compared to HS. Comparing both patient groups, PD-MED–DBS revealed higher anterior COP displacements than PD-MED (Figure 5).

Both PD-MED and PD-MED–DBS showed longer motion times with slower velocities in the anterior and the posterior directions compared to HS (Table 2). No statistically significant differences were found comparing PD-MED and PD-MED–DBS. The overall coefficient of variation showed higher group variability for PD-MED (0.38 ± 0.16) compared to those of PD-MED–DBS (0.32 ± 0.14) and HS (0.30 ± 0.16).

Intra-Group Comparisons

Within groups, no significant differences were found between male and female subjects or between patients with right-sided or left-sided disease dominance. However, various statistically significant intra-group differences (*p* < 0.004) between the anterior and the posterior directions were found for each group. Predominantly, COP displacements and COP mean motion velocities showed higher values for the anterior direction compared to the posterior direction.

#### 3.2.3. Gait

Inter-Group Comparisons

Statistically significant differences for gait parameters were mainly found between both patient groups and the group of healthy subjects. Patients in PD-MED and PD-MED–DBS walked significantly slower with less stride length and longer stride and double the support times compared to those of HS. PD-MED took significantly fewer steps per minute with smaller COP lengths compared to HS. Both patient groups showed broader stride widths compared to HS; however, this was only statistically significant in PD-MED–DBS. No differences were found when comparing PD-MED and PD-MED–DBS (Table 3). The overall coefficient of variation for the gait parameters showed higher variability for PD-MED (0.20 ± 0.06) than that of PD-MED–DBS (0.15 ± 0.07) and HS (0.12 ± 0.05).

Intra-Group Comparisons

Within groups, no significant differences were found between male and female subjects or between patients with left-sided and right-sided disease dominance. There were also no differences for gait parameters between the left and right body sides. 

### 3.3. Plantar Cutaneous Vibration Perception

Inter-Group Comparisons

No considerable differences between groups were found for contact forces or plantar temperatures. As shown in Figure 6, only the left metatarsal head of PD-MED showed statistically significant differences for the VPTs compared to HS. Nevertheless, patient VPTs seemed to be higher in general compared to VPTs from HS. Although there were no statistically significant differences between patient groups, VPTs from PD-MED tended to be higher compared to the VPTs from PD-MED–DBS. The overall coefficient of variation for VPTs showed slightly higher variability for HS (0.25 ± 0.05) compared to PD-MED (0.18 ± 0.02) and PD-MED–DBS (0.19 ± 0.03).

Intra-Group Comparisons

In all three groups, women had lower VPTs than men: however, these were only statistically significant for PD-MED. In PD-MED, the VPTs of the left foot were significantly higher compared to the VPTs of the right foot. Moreover, an effect of the disease-dominant side was found but was not consistent throughout the two patient groups. For instance, patients in PD-MED with left-sided disease dominance showed higher VPTs for the left foot (46.7 µm ± 24.4 µm) compared to the right foot (34.9 µm ± 20.5 µm). Nevertheless, this effect was not found for patients in PD-MED with right-sided disease dominance or for patients in PD-MED–DBS.

## 4. Discussion

### 4.1. Motor Performance

Our first objective was to analyze patients’ axial motor symptoms, such as postural instability and gait difficulties, in comparison to healthy elderly subjects. We also analyzed whether patients treated with anti-parkinsonian medication in conjunction with STN-DBS had superior motor control compared to patients treated with medication alone. Therefore, we implemented different motor tests and used a pressure distribution platform to objectively assess subjects’ motor performance.

#### 4.1.1. Quiescent Bipedal Stance

Adequately functioning postural control is the foundation on which motor tasks are executed, such as quiescent upright bipedal stance [8,23,119]. As expected, the stance of both patient groups PD-MED and PD-MED–DBS was unstable and jerky compared to the healthy subject group HS. In more detail, patients showed higher COP sway displacements and higher COP sway velocities, which is consistent with several other studies and indicates that, even in situations with low balance requirements, PD patients have diminished postural control [12,13,20,21,23,26,44,103,104,105,120,121,122,123,124,125,126,127,128,129]. However, as shown by several other studies, patients’ postural instability during quiescent stance does not always manifest with increased COP values but can also present as close to normal or even smaller values [12,28,130,131,132]. This might especially be the case for the akinetic-rigid PD subtype characterized by increased background muscle activity, which leads to a higher degree of co-contraction of the leg muscles and higher joint stiffness [133,134,135]. Consequently, akinetic-rigid PD patients might resort to joint stiffness as a balance strategy because it increases movement resistance and consequently diminishes sway amplitudes [133,134,135,136,137]. In our study, however, the patient groups comprised various symptom mixtures of the main PD subtypes: akinetic-rigid, tremor-dominant, postural instability, and gait disturbance. The overall increased COP values we found for both patient groups might also be explained by the fact that even tremor can interfere with postural sway [138].

Despite their postural instability, the patients in this study did not choose wider stances than healthy subjects, which would have led to more stability due to an increased base of support. One explanation for this might be that axial rigidity and bradykinesia can influence patients’ standing positions [27,133]. In this regard, it has been shown that PD patients have inadequate anticipatory postural adjustments prior to self-initiated movements and that they have difficulties scaling them up, as needed with a wider stance. Therefore, it may become a subconscious habit for PD patients to stand with a narrow stance, similar to healthy subjects, even though this motor test did not require any movement initiation [26,27,133,139]. Another explanation why patients did not compensate for their unstable and jerky stance with a wider standing position might simply have to do with the low level of difficulty and little requirement for balance control necessary to execute bipedal quiescent stance. Even though postural instability is noticeable in early stages of the disease (Hoehn and Yahr Stage 2) and our patients showed significantly lower self-rated balance confidence, bipedal quiescent stance as a daily activity still seems to be within the balance capacity of the PD patients tested in our study [12,23,140,141,142]. 

Among the investigated COP parameters, we found that spatial parameters, such as COP ranges and the COP 95% confidence area, seem to be the most promising variables to discriminate between patients and healthy subjects. As supported by De la Casa-Fages et al., the COP 95% confidence area showed a specificity of 70.6% and a sensitivity of 100% for differentiating between PD patients and healthy subjects [143]. In this respect, the COP velocity has also been mentioned as a valuable parameter [20,28]. 

On the other hand, there seems to be no consensus as to which body direction might be more affected by postural instability. While some authors argue that the ML direction is most impaired [20,129,144], others claim that patients are predominantly prone to falls in the AP direction [12,28,44,135,145]. In our study, we found that PD patients have pronounced postural instability in both the ML and AP directions compared to healthy subjects. However, comparing between the two directions within the study groups, the AP directions seem to be more instable than the ML direction in general, even for the healthy subjects. There are basically two different postural strategies that control sway in AP and ML directions: the hip strategy and the ankle strategy [21,133,146,147]. Hip adductor and abductor muscles are mostly involved in controlling postural sway in the ML direction, whereas postural sway in the AP direction involves the activity of plantarflexor and dorsiflexor muscles of the ankle joints [20,28,148]. The direction-specific postural instability in PD patients may result from neural impairments of the disease. Hence, patients’ poor muscular control and lower limb coordination caused by higher background activity and related muscle fatigue might be some factors of the jerky and instable balance performance in both directions, especially in the AP direction [12,27,106,149,150]. Nevertheless, both directions seem to be good discriminators between PD patients and healthy subjects [129]. 

The postural instability found in both of our patient groups might reflect the underlying pathophysiological mechanisms of the disease. Nevertheless, those mechanisms are complex and intricate, as they include numerous neurological structures and their functional interaction. However, the basal ganglia play a key role [4,151,152,153]. It is generally known that the decline of dopaminergic neurons within the substantia nigra pars compacta results in a hypo-dopaminergic state within the basal ganglia. Insufficient dopamine levels create imbalance in the excitatory and inhibitory communication within the entire network of the nigro-striatal and thalami-cortical circuits [5,6,7,154]. This includes structures such as the basal ganglia, the thalamus, the primary and premotor cortices, the supplementary motor area, the visual area of the frontal lobes, the intermediate cingulate cortex, the brainstem, the reticulospinal system, and the cerebellar vermis of the cerebellum, to name a few [4,5,155,156]. Any abnormalities disrupting the interaction of those structures create obstacles for selecting, planning, adapting, and executing motor functions. Although the primary pathophysiological feature of the disease has traditionally been attributed to the deficiency of dopamine, there is also evidence suggesting that motor symptoms in Parkinson’s disease can be exacerbated by the dysfunction of non-dopaminergic circuits, including the cholinergic system [157,158,159,160]. Studies have shown that the reduced activity of structures that compose the mesencephalic locomotor region, such as the pedunculopontine nucleus and its thalamic projections, as well as the adjacent cuneiform nucleus are associated with axial motor symptoms in PD [157,160,161,162,163,164]. The negative multisystem hypo-cholinergic effect on motor performance seems to be additionally intensified by the denervation of the basal forebrain [23,157,165]. 

When comparing PD-MED with PD-MED–DBS, we could not find substantial differences for postural performance during the quiescent stance test. This seems surprising, since numerous studies have reported superior effects of DBS over medication for postural control in general [8,12,15,22,121,123,125,128,129,143,166,167,168,169,170,171,172]. However, there are also studies that report lagging beneficial effects of STN-DBS on postural control, especially in conjunction with medication, which may support our findings for this specific balance test [26,143,168,173,174,175,176]. For instance, studies by Yin et al. and De la Casa-Fages et al. indicated that STN-DBS demonstrated improvement in motor performance without the use of medication [143,168]. McNeely and Earhart observed no synergistic effect of combining medication and STN-DBS, and according to Rocchi et al., DBS might have mitigated the adverse effects on motor performance induced by levodopa [176,177]. 

The heterogeneity between studies might be owed mainly to methodological differences, such as studies using more subjective clinical scales instead of computerized methods, or different parameters and testing conditions, as well as different group compositions [13,23,128,178]. In addition, the influence of varying disease severity and time-related fading of DBS-induced benefits might have had negative effects on our results [98,99,100,171,179,180]. Another explanation might have to do with the low level of difficulty of the quiescent bipedal stance test. As already suggested, this test might not have exploited patients’ balance capacities as needed to eventually differentiate between the two patient groups. In this respect, Benatru et al. and Błaszczyk et al. concluded that quasi-static posturography in PD might have limited sensitivity, because it only assesses a single component of balance and therefore should be supplemented by additional, more challenging measures of the stability range, such as functional reach or maximal voluntary leaning tests [12,142].

#### 4.1.2. Functional Limits of Stability

The functional limits of stability test are known to be a valuable application-oriented measure of dynamic stability to assess patients’ risk of falling [20,104,105,181]. Within a controlled and safe scenario, the test simulates challenging daily balance situations such as the transition from sitting to standing, the initiation of gait, or leaning and reaching out to grab something [104,106,181,182,183]. Since PD patients are mostly confronted with self-generated internal balance perturbations, this test might be a sensitive and objective measure with potential to distinguish between PD patients and healthy subjects [103,105,106,148,182,183,184,185], specific PD subtypes [20,105,184,185], and therapy-induced effects on postural control [15,104,169,186,187]. In the current study, this test evaluated subjects’ ability to voluntarily lean as far as possible during standing without losing balance. Therefore, we analyzed the maximum COP displacements and the motion times and velocities during leaning in the anterior and posterior directions. 

For the interpretation of maximum COP displacements, it is important to analyze the neutral COP location during upright stance, because any shift in the neutral COP location towards a specific direction influences the capabilities of voluntary maximum leaning. In this regard, studies have shown that concomitant parkinsonian symptoms can shift the neutral COP location during upright stance. For example, Schlenstedt et al. found that the COP location was situated more posteriorly in PD patients with distinct signs of freezing, which might have been a compensatory strategy to obtain a safe stance position and avoid falling forward [105]. The authors of another study reported that PD patients with pronounced camptocormia and stooped posture had a significant COP shift towards the anterior compared to healthy subjects [26,142,188]. As our study groups did not comprise patients with severe freezing or camptocormia, the neutral COP position during upright standing was comparable between all three groups and was located at approx. 40% of the foot length measured from the heel. Mancini et al. also found comparable COP locations between healthy subjects and PD patients during quiescent stance, which suggests better comparability between study groups for the functional limits of the stability test [104].

As we found during forward and backward leaning, both of our patient groups showed significantly reduced total COP ranges due to smaller displacements in the anterior and posterior directions compared to our healthy subject group. In other words, PD patients had significantly diminished limits of stability in the sagittal plane and therefore had much less functional balance reserves in which they could safely sway without falling or being forced to execute compensatory steps [26,104,183]. Similar results have also been found by several other studies; however, there is inconsistency regarding the leaning direction in which PD patients have greater balance restrictions [103,104,105,106,148,182,183,184,185,187]. For example, using force plates, Nikaido et al. and Dona et al. also found that PD patients’ functional balance reserve values in the sagittal plane were smaller compared to those of healthy subjects [103,148]. In an older study, Schiepatti et al. reported that during forward and backward leaning, the total COP range in the anterior–posterior direction was approx. 40% of the foot length for healthy subjects and approx. 30% for PD patients. This, however, is much less compared to our study findings [183]. More comparable values were found in the study by Mancini et al., in which healthy subjects showed a total COP range in the sagittal plane of approx. 50% of the foot length and PD patients showed approx. 40%. However, in contrast to our study findings, the authors concluded that patients’ limits of stability were primarily a result of the reduction in the maximum body leaning forward, but not backward [104,105,122].

Besides their restricted limits of stability, our PD patients also required more time to reach their maximum leaning posture and consequently showed slower velocities compared to the healthy subjects. When comparing between forward and backward leaning, our patients required significantly longer times to reach the maximum posterior COP displacement. These results are supported by Mancini et al. and Kelly et al. Nevertheless, in Mancini’s study, patients only showed slower velocities during backward leaning [104,185].

There are several factors that might explain patients’ diminished limits of stability and slow movements during maximal voluntary leaning. Physiological factors, such as bradykinesia and rigidity, may have a strong contribution, since they lead to diminished joint mobility, axial inflexibility, and muscle weakness [142,184,189,190,191]. Due to the influence of axial rigidity, it has been reported that PD patients had less postural sway compared to healthy subjects while holding the critical inclined posture [28,183,192]. The authors concluded that the pronounced oscillating body sway in healthy subjects at the boundaries of stability might serve as a balance strategy, whereby motor control continuously changes between keeping balance (actively opposing gravity) and letting go (complying with gravity) [183]. Hence, PD patients may struggle to achieve and sustain greater COP displacements due to compromised temporal discrimination. This impairment could prevent the execution of the alternating minor sway movements that are typically performed by healthy subjects [183,193,194]. Slowness of backward leaning may especially reflect patients’ weakness in the ankle extensors [104]. Another physiological factor includes somatosensory integration for regulating motor control. Parkinsonian motor deficits, such as rigidity and bradykinesia, are generally not isolated deficits, but strongly associated with impaired integration and processing of afferent sensory information from the visual [20,21,22,23], vestibular [8,24,25,26], and somatosensory [29,30,195] systems. Postural control is a closed-loop circuit, which integrates afferent inputs for controlling muscular agonists (activation) and antagonists (inhibition) through direct and indirect pathways of the basal ganglia [151,153,196,197,198]. In fact, the ability to use sensory information seems to depend on the stage of striatal dopamine loss, suggesting the role of dopaminergic pathways for central sensory integration [28,199]. Due to the inclined posture while performing the functional limits of stability test, balance control relies particularly on the vestibular and somatosensory systems. The vestibular system provides information about the position of the body in space and the relative position of each joint with respect to other parts of the body [24,39,40]. In PD patients, it has been reported that vestibular hypofunction and impaired signal processing in higher-order sensory integration centers might cause impaired sense of trunk position and tilt misperceptions [8,24,25,27,28]. In addition, the functionality of somatosensory sensors, such as Golgi tendon organs, muscle spindles, joint receptors of the lower extremities, and plantar cutaneous mechanoreceptors, is important for leaning the body towards the limits of stability. These sensory systems provide valuable information about muscle contraction status and ligament and tendon tension [29,30,35,42,195,196,198]. Plantar cutaneous mechanoreceptors gather important information about the pressure distribution and loading shifts underneath the foot during movements and are therefore heavily involved in adapting muscle contraction tone and contraction patterns for reaching the maximal COP displacements [41,42,43,44,45,46,47,48,49,50,51,52]. In PD, all of these afferent sensors show deficits that could have prevented patients from reaching higher limits of stability. We note that the functionality of specific plantar cutaneous mechanoreceptors and therapy-induced effects in PD is another main focus of this study which is discussed later in more detail. There are also psychological factors that might explain our results. This test is considered to measure “functional” limits of stability because patients’ postural performance also depends on subjective perception, internal postural control abilities, and fear of falling [20,104,186,200]. To minimize the weight of fear, our patients were assured that an examiner who was standing nearby during the test would keep them from falling. Nevertheless, as our patients also showed lower self-rated balance confidence values compared to the healthy subjects, fear of falling still might have played a major role in preventing patients from leaning further. Slowness during leaning may also be related to cautiousness and fear of falling [200].

Regarding the effects of DBS and medication, group PD-MED-DBS, which received both therapies simultaneously, performed overall better compared to PD-MED, which was treated with medication alone. More specifically, PD-MED–DBS showed larger limits of stability due to significantly higher COP displacements in the anterior direction. Furthermore, group PD-MED–DBS showed a strong trend towards more normalized values for the COP motion times and velocities in both leaning directions compared to group PD-MED. Even though the literature reports divergent results for the effects of each therapy on the functional limits of stability, better overall effects seem to be achieved with deep brain stimulation alone or in conjunction with medication, which supports our study findings [15,169,186,187]. For instance, Haito Li et al. showed that balance performance began to improve at 6 months post-surgery DBS, with significant improvements in patients’ limits of stability [169,186]. May et al. and St. George et al. showed improved abilities to reach and lean as far as possible after stimulation was turned on [15,187]. The authors suggest that STN-DBS may enhance the central integration of afferent sensor information by regulating the activity of neurons connected to STN. Hence, the normalized functionality of the entire cortical–striatum–pallid–thalamic–cortical system loop may improve sensory–motor interaction and enable patients to maintain balance, even during inclined posture [4,186,201,202,203,204]. 

The limited efficacy of dopaminergic medication could be explained by considering that motor dysfunction in PD is not only caused by the depletion of the dopaminergic system but also involves the cholinergic systems. Therefore, dopaminergic medication may relieve dysfunction associated with dopamine but does not influence motor dysfunction related to the cholinergic system [143,160,205]. Furthermore, axial motor symptoms progressively become resistant to medical treatment over time, and adapting to consistently higher doses of medication may exacerbate additional balance-related disabilities [98,161,203,206]. Nevertheless, the literature shows conflicting results for improvement [104,185] and no improvement [15,169] of the functional limits of stability after medication intake. 

Similar to our study, DBS therapies and levodopa medication are generally used in combination, contributing to positive synergistic effects on postural performance. For example, Bejjani et al. discovered that the combination of STN-DBS and levodopa enhanced total motor ability by approx. 80%, in contrast to supra-threshold doses of levodopa alone [207]. Colnat-Coulbois et al. concluded that the combination of STN stimulation and levodopa treatment facilitated basal ganglia functionality. This might be attributed to the separate influence of stimulation on both dopaminergic and non-dopaminergic pathways [172]. Lubik et al. also concluded that the combination of both therapeutic methods showed the best motor outcome [208]. Reducing medication dosages in conjunction with DBS might represent another crucial factor contributing to synergistic effects, as it diminishes side effects induced by levodopa [209,210]. Nevertheless, there are also contradictory findings that do not support the hypothesis of synergistic effects of STN-DBS and medication [143,168,174,175,176,177]. However, studies investigating synergistic effects on the limits of stability are rare and use rather subjective clinical scores. There are two studies showing positive effects using both therapies in tandem [15,187]. May et al. suggested that both treatment strategies improved participants’ abilities to reach and lean to the limits of stability. Furthermore, St. George et al. reported synergistic effects suggesting that combining DBS and levodopa medication improved balance abilities beyond what either treatment provided alone [187]. In this regard, there are also studies that show that reducing medication-induced complications, such as dyskinesia and symptom fluctuations through DBS, might enhance patients’ balance performance [158,166,186,209,211]. 

The discrepancies between studies might mainly have methodological reasons, such as different independent group compositions with varying disease severities and stimulation durations. Moreover, varying executions of the functional limits of stability test and not normalizing the maximal COP displacement to the length of the feet might have added bias to the results. 

#### 4.1.3. Gait

The parkinsonian gait pattern is one of the most typical hallmark symptoms of the disease, and generally manifests as a slow and unsteady gait, with shortened and shuffling strides. To characterize patient gait pattern more specifically and determine which parameters show the most clinical value in terms of differentiating between PD patients and healthy subjects, as well as therapy-induced differences between medication and medication in conjunction with DBS, we used a pressure distribution platform and analyzed various gait parameters.

As a spatial–temporal parameter, gait velocity depends on the stride length and the stride time or the rate at which a person walks [212,213]. As our study revealed, the fundamental disturbance of patients’ slowness of gait seems to be primarily a result of a significantly reduced stride length rather than reduction in cadence. Our results are therefore in agreement with most other study findings [103,208,212,213,214,215,216,217]. In a meta-analysis including more than 50 studies, Zanardi et al. showed that stride length in PD is on average approx. 16 cm shorter than it is for healthy subjects, whereas cadence is on average only approx. 1.75 steps higher than it is for healthy subjects [213]. Due to the reduction in stride length, both of our patient groups only reached an average self-selected gait speed of approx. 0.8 m/s. According to Roper et al., this indicates ‘‘limitations in community mobility’’ [214]. Patients’ reduced stride length might furthermore reflect the shortened lengths of the COP during the stance phase of gait. With shorter strides, the initial and terminal contact points of the foot with the ground are both shifted towards the middle of the foot, which consequently results in shorter COP lengths for PD patients during walking.

Another parameter indicating patients’ reduced quality of gait is stride width. In contrast to our findings for the quiescent bipedal stance test during which patients and healthy subjects had the same narrow foot positions, patients had a broader gait than our healthy subjects. As stride width generally enlarges the base of support, this could be interpreted as a compensation mechanism through which patients try to counteract disease-related motor difficulties that might impair muscle coordination and stability [27,213,218,219,220]. During gait initiation, it has been shown that the first steps tend to be wider in PD [220]. Since gait velocity changes lower limb kinematics, patients’ broad gait might also be an effect of slower walking with shortened strides. Due to patients‘ diminished forward walking progression, their limbs are aligned more broadly to compensate for medio-lateral instability [221]. Nevertheless, this is surprising since most other studies, including a meta-analysis, found gait width in PD patients to be similar to or even narrower than it is in healthy subjects [213,217,222]. Therefore, it has been suggested that patients’ compensatory gait broadness may be overcome by the interference of rigidity, which might impair the hip abductors and therefore affect the range of transversal hip rotation and pelvic tilt [218,219,223]. 

A good measure of instability during gait is the double support time of the gait cycle, which is defined as the period of time when both feet are in contact with the ground [208,213,224,225]. In our study, PD patients in both groups were in the double support phase significantly longer than the healthy subjects, which suggests reduced dynamic gait stability. This is supported by a meta-analysis by Zenardi et al., who found that PD patients’ double support times were approx. 1.79% longer on average compared to those of healthy subjects [213]. Longer double support times can be attributed to an inability to adequately transfer weight to the standing foot, as well as to reduced ankle push-off, which relates to rigidity and muscle weakness [226,227,228]. Since most gait parameters are closely related, the elongated double support time might also be a result of patients’ reduced walking speed, reduced stride length, and changes in cadence [212,227]. 

Interestingly, those distinct gait difficulties that we found using the pressure distribution platform were poorly reflected by the values of the clinical rating scales. For example, our patients’ average disease severity was approx. two on the Hoehn and Yahr scale. This is a value that excludes impairments of balance or gait. However, the mismatch between our findings and the clinical ratings might be explained by the influence of subjectivity and the coarse scoring of most clinical rating scales [23,229]. Moreover, the tests in our study were carried out several months after the clinical examination, which could have emphasized this discrepancy due to disease progression [171,214].

The parkinsonian gait pattern with reduced velocity and shortened strides might be determined predominantly by rigidity and bradykinesia [170,173,208,212,230,231]. The presence of gait festination, which is the execution of quick and small strides to prevent patients from losing balance, could also contribute to the manifestation of this pattern [212,215]. Patients’ hypokinetic gait motion with reduced floor clearance relates to altered muscle activity with pronounced co-contraction and muscle weakness, which also reflects a higher metabolic effort for PD patients during walking [127,141,171,213,215,216,228]. Decreased arm swing amplitudes and chaotic arm swing timing during gait might also impede patients’ gait propulsion [12,135,145,232]. Since motor control is dependent on functional afferent sensory systems, disease-related deficits of the visual [20,21,22,23], vestibular [8,24,25], and somatosensory [29,30,42,195] systems may be another contributing factor to the parkinsonian gait pattern. Psychological factors contributing to patients’ slow gait might be predominantly fear of falling [149,233,234]. This is supported by the significantly lower self-rated gait confidence levels compared to those of our healthy subjects. 

There is no easy answer as to what causes the underlying neurological pathophysiology of the parkinsonian gait pattern, mainly due to the intricate interaction between multiple structures of the central nervous system that are responsible for gait regulation [4,5,6,151,154,155,156,157,159,160,201]. Nevertheless, several mechanisms have been suggested in the literature. In addition to disturbances emanating from the basal ganglia, it has been proposed that degeneration of cholinergic neurons in the basal forebrain is a significant contributor to gait disturbance in PD [157,165,235]. Hence, PD patients’ decreased gait speed might be an early indicator of basal forebrain cholinergic deficits [157]. With advanced imaging technologies and microelectrode recordings, it has also been demonstrated that the mesencephalic locomotor region plays a crucial role in the pathophysiology of parkinsonian gait [236,237,238]. Since specific areas within the pedunculopontine nucleus comprise approx. 40% of neurons responsible for motor control, their degeneration seems significant [157,160,162,163,164,238,239]. Hence, disrupted afferent connections between the pedunculopontine nucleus, the motor cortex, and the basal ganglia may lead to an inability to control stepping mechanisms [212,240,241,242]. Gait disturbance in PD might also be related to hyperactivation of the cerebellum, a structure known to be crucial for motor coordination [215,236,243]. Cerebellar hyperactivation is considered to be an adaptive response of the central nervous system to compensate for the impaired functionality of the basal ganglia and the brainstem [215,243]. Furthermore, patients’ disorganized movements and hypokinesia may be attributed to increased inhibition of thalamo-cortical fibers, caused by hyperactivity of the globus pallidus internus [127,244]. While interrupted supra-spinal communication between the phasic output from the basal ganglia and the supplementary motor area may explain impaired stride length regulation, defective mechanisms at the spinal level and the brainstem could play a crucial role in the dysregulation of cadence [212,216,241,245,246]. 

The multisystem degeneration of cholinergic structures, which are responsible for movement control, may explain the increasing dopamine non-responsiveness of gait difficulties in PD. DBS, which acts on cholinergic systems, partially restoring the functionality of the dopaminergic system, and therefore improving dopamine-responsive symptoms, might help explain why DBS generally performs better than medication in treating parkinsonian gait disorders. Although the literature is controversial, there are various studies that report beneficial effects of DBS on parkinsonian gait [127,167,170,171,176,208,212,214,216,224,225,230,247,248,249,250,251]. In summary, most studies indicate an enhanced stride length that contributes to increased speed, while the cadence is usually unaffected by DBS. As a general rule, the effect of DBS on gait parameters should be at least as effective as supra-threshold doses of levodopa, but it is likely even more effective [208,212,214,216,251,252]. For example, using a pressure-sensitive treadmill, Navratilova et al. found increased step lengths and reduced double support times 3 months after starting neuro-stimulation. The authors concluded that DBS treatment has a superior impact on gait compared to medication alone [225]. This is supported by the results of a meta-analysis by Roper et al., who also reported improved gait speed after switching on DBS [214]. Faist et al. reported that patients’ stride length and mean walking velocity increased almost threefold with STN stimulation, whereas levodopa alone showed smaller effects [212]. Johnsen et al. and Lubik et al. found improved gait performance with increased velocity and stride length comparing the on- vs. the off-stimulation states. The authors of both studies also reported improved balance during gait, since patients in the on-stimulation state spent less time in the double support phase [208,224]. Lubik et al. however, reported that medication led to a more prominent increase in step length than DBS, whereby the increase in cadence was most pronounced with DBS [208]. Mera et al. and Hausdorff et al. reported reduced variability of gait parameters with DBS, which suggests enhanced gait coordination and stability [249,251,253].

As DBS is usually accompanied by medication, our question was whether both treatments in combination have a positive synergistic effect on gait performance in PD. According to the literature, reports about synergistic effects are inconsistent. For example, using clinical scales such as the UPDRS and the gait and balance scale, it has been reported that gait performance improved significantly more when patients were simultaneously in the on-stimulation and on-medication states [207,208,247,251]. Using computerized methods, Hausdorff et al., Lubik et al. and Faist et al. reported, more specifically, that the combination of both therapeutic treatments showed the best results on gait parameters, including gait speed, stride length, and cadence, as well as stride-to-stride variability [208,212,251]. Stolze et al. reported that both treatments together further increased gait velocity by approx. 30% compared to DBS alone [216]. However, such synergistic effects were questioned by the findings of various other studies, which failed to observe further improvements in spatial–temporal gait parameters when DBS was added to chronic daily dopaminergic therapy [175,176,177,254,255]. In our study, although the differences between the two patient groups were subtle, the patients who received both treatments, medication and DBS, simultaneously generally showed better gait performance compared to the patients treated with medication alone. Parameters that indicate better gait performance in PD-MED—DBS include stride length, cadence, double support times, and the COP length. Moreover, the overall coefficient of variation for gait parameters was smaller for PD-MED—DBS compared to PD-MED, which indicates better gait control. Therefore, our findings support those studies that reported beneficial synergistic effects. 

Improvements in gait parameters using DBS are speculated to be secondary, mainly due to diminished rigidity, bradykinesia, and dyskinesia [170,171,249,251,252,256,257]. Therefore, enlarged stride length and higher gait velocity are the results of enhanced range of motion of the ankle, knee, and hip joints, as well as increased arm swing amplitudes [171,212,216,249,251,254,256,257,258,259]. In addition, improved vertical alignment of the trunk and shank with reduced trunk forward inclination and increased trunk mobility in terms of lateral bending and torsion range may be positive factors [120,125,188,252,256,260,261,262]. DBS-induced gait improvements might also be associated with enhanced lower limb muscle strength, the normalization of leg muscle contraction patterns, and inter-limb coordination [125,141,167,212,226,252,255,259,262,263,264]. As hypothesized in this study, the normalization of proprioceptive functionality and enhanced processing and sensory–motor integration of afferent information through DBS may have additionally caused enhanced gait performance [19,32,33,34,36,74,90,91,265,266]. 

The exact mechanisms of DBS and its synergistic effects on gait in combination with medication are still not fully understood. However, due to modern functional imaging technologies, it has been proposed that DBS affects not only a single location, but has combined effects at multiple levels simultaneously, including the cortex, the brainstem, and circuits originating from the diencephalon [240,267]. Several studies suggest DBS inducing modulation effects rather than an exclusive inhibition of neurons or axons in the stimulated area [202,224,268,269,270]. More precisely, DBS of the STN might improve gait by modulating cortical areas that are associated with the preparation and execution of movements. Moreover, STN-DBS may modulate pallido-nigrofugal projections to brainstem regions responsible for generating locomotor patterns [240,267]. Other studies report that the disrupted functional interaction between STN and the sensory–motor and fronto-parietal cortical regions in PD patients with freezing of gait might be improved directly through the stimulation effect of STN on the cortico–striato–pallido–thalamo-cortical system [170,271,272,273]. Investigations with positron emission tomography of regional cerebral blood flow and metabolism indicate increased activity in the STN during stimulation and thus activity of the output nuclei globus pallidus interna and pars reticulata of substantia nigra [274,275]. Another concept includes DBS-induced normalization of the firing rate of the pedunculopontine nucleus [157,159,161,162,207,208,238]. Synergistic effects, in which the efficacy of levodopa was potentiated in the presence of DBS, may be related to different but additive effects of both treatments on the brain’s central pattern generators or on the pedunculopontine nucleus, which are structures that help regulate gait rhythmicity and stability [212,238,251,263,276,277]. Additional improvements in gait resulting from STN-DBS and medication may also be attributed to improvements in abnormal cortical activity. This involves the interaction between the supplementary motor area and the basal ganglia, which are crucial for movement preparation [176,255,278,279]. DBS-induced reductions in medication dosages, which in turn diminish medication-related side effects such as troublesome dyskinesia, might be another major factor facilitating synergistic effects [209,210,230,280].

The subtle positive effects of DBS on gait found in our study could have several reasons. For example, we presume that disease progression was an issue. This is supported by our findings that patients in group PD-MED–DBS were affected by the disease approx. 2.2 times longer, on average, than those in PD-MED. Because of the relatively long stimulation intervals of approx. 3 years since surgery, fading of DBS-induced benefits might also have had negative effects [98,99,100,171,179,180]. For example, Brozova et al. observed positive effects in patients who were treated within 1–2 years, and a significant decline in patients who were treated for 5 years or more. Analyzing results of 11 studies in their meta-analysis, St. George et al. found that gait performance progressively declined after as little as 2 years and was even worse compared to the pre-surgery state [230]. As suggested, fading of stimulation-induced benefits may be associated not only with a natural progression of PD symptoms but also with the progressive interference of non-dopaminergic mechanisms that are not responsive to DBS [98,99,161,171,230,247,281]. Moreover, the lack of DBS efficacy can also be assigned to suboptimal stimulation settings [171]. As some of our DBS patients underwent testing before their neurological consultations, they might not have experienced the benefits of optimized DBS settings. This potential bias could have influenced our results towards lower efficacy. A bias in the results and limitations in the interpretability of our findings could have additionally been introduced by the lack of a more precise classification of specific Parkinson’s disease subtypes, as well as the absence of testing patients in the “off” state condition of both therapies.

### 4.2. Plantar Cutaneous Vibration Perception 

Our second objective was to investigate tactile somatosensory functionality of the plantar skin in PD patients in comparison to healthy subjects. We also analyzed the therapeutic effects of whether anti-parkinsonian medication alone or medication in conjunction with DBS show different effects on tactile somatosensory functionality of the plantar skin in PD patients. Therefore, we implemented a customized vibration exciter to analyze subjects’ plantar cutaneous vibration perception thresholds. 

We hypothesized that PD patients have higher plantar cutaneous vibration perception thresholds than healthy subjects, which was confirmed by our results. Both of our patient groups, PD-MED and PD-MED–DBS, showed generally higher thresholds; however, statistical significance was only found between PD-MED and HS. Although investigations about tactile cutaneous perception of vibration or pressure in PD are rare, especially for the foot sole, there are a few studies which help to explain our results. Compliance with our study findings can be found in three other studies [64,87,88], while two other studies failed to find PD affecting cutaneous thresholds for mechanical stimuli of the foot [48,89]. Testing cutaneous sensory functionality and analyzing skin biopsies of the foot, Nolano et al. showed that PD patients have significantly increased tactile thresholds, which is strongly associated with patients’ significant loss of epidermal nerve fibers and mechanoreceptors, such as Meissner corpuscles [64]. In another study, Prätorius et al. found significantly higher thresholds in PD patients when analyzing five sites of the plantar foot using a vibration exciter at 30 Hz and Semmes-Weinstein Monofilaments for touch pressure perception. Their results showed that for each tested location (except the heel), the thresholds of PD patients were at least twice as high as those in the healthy control subjects [88]. Using electrical sinusoidal stimulation at 5, 250, and 2000 Hz at the external malleolus of the foot, Ikeda et al. also found significantly increased perception thresholds for PD patients compared to those of healthy controls [87]. However, McKeown et al. reported intact cutaneous functionality in PD. Using sophisticated methods, the authors investigated vibro-tactile thresholds at 30 and 250 Hz at the first metatarsal head of the foot and failed to find evidence of elevated plantar thresholds in PD patients [48]. Doty et al. also found no impaired tactile pressure sensitivity in PD patients at the medial sole of the foot and the plantar halluces using a forced-choice staircase threshold test paradigm [89]. The lack of agreement between different study results might primarily be attributed to varying methodological factors, such as the different severity of the disease between the study groups investigated. In this regard, Nolano et al. found that disease severity correlates with the loss of epidermal nerve fibers and Meissner corpuscles in PD. Hence, disease severity, which is associated with disease duration, might be a valuable factor influencing tactile perception [64]. However, when comparing the findings of different studies, there appears to be no consistent pattern for the influence of disease severity and duration on tactile cutaneous perception. In other words, even in studies in which patients had suffered from PD for a relatively long period or had a higher severity of the disease at the time of measurement, no differences in tactile perception were found between PD patients and healthy subjects [48,89]. As sensory perception underlies multifactorial influences, further factors, such as different measurement devices, testing other anatomical locations, and varying vibration frequencies, might also play a role. Furthermore, since only the current study and that by McKeown et al. controlled external factors, such as contact force, this could be another factor influencing tactile perception thresholds [48]. In addition, potential differences in patient skin temperature, which was only controlled in our study, might have had an unknown effect on tactile perception in other studies. Since our healthy subjects were slightly older compared to our patient groups, this could have biased their VPTs towards higher values as an effect of aging. Therefore, study groups with comparable age would have shown differences more dramatically. Finally, the lack of statistical power due to small sample sizes might have contributed to different results as well. However, in addition to studies that investigated tactile perception of the foot, there are a number of studies that analyzed tactile perception at other anatomical locations in PD, including the arms, hands, fingers, and the torso [32,33,34,36,37,38,81,90]. Most of these studies found that PD increases tactile perception thresholds, which confirms the results of our study. 

Given that only the soles of the feet are in direct contact with the ground while standing and walking, afferent information from plantar cutaneous mechanoreceptors is crucial input for motor control. Plantar cutaneous mechanoreceptors gather information about the pressure distribution and load shifts underneath the foot during movements, and therefore are involved in adapting muscle contraction tone and contraction patterns. Sensorimotor integration of plantar mechanoreceptors has already been investigated in several studies with individuals without neurological disease. In summary, decreased plantar cutaneous sensation achieved by anesthesia of the foot sole leads to impaired control of static and dynamic balance abilities [42,46,49,50,51,52,53,54,55,56]. Conversely, sensory stimulation of the foot sole has been shown to improve balance and gait performance. This has been demonstrated in several studies with PD patients using various types of shoe insoles. For instance, Phuenpathom et al. analyzed the effect of mechanical pressure stimulation of the foot sole during the initiation of gait in PD patients. They used insoles with thickened silicon pads and found that the pressure stimulation of the plantar foot skin reduced the freezing of gait, which is a devastating motor symptom in PD [43]. In another study, Qiu et al. reported that textured insoles decreased postural sway and improved balance stability under challenging conditions in the PD group due to facilitating afferent information from mechanoreceptors of the foot sole [44]. Another study analyzed predicting factors of falls in PD. Investigating muscle activity and spatial–temporal parameters during walking, Jenkins et al. found an improvement in the overall stability and safety of gait when using stimulating ribbed insoles. More specifically, the authors reported increased single-limb support time, which implies improved overall stability and normalized timing of the peak activation of the tibialis anterior muscles [45]. Novak and Novak used an elastic vibrating insole that delivered 70 Hz suprathreshold vibration bursts to the heel and the forefoot during the stance phase of gait. Their findings indicate that step-synchronized vibration stimulation of the plantar foot improves gait steadiness in PD with predominant balance impairment. Suprathreshold vibration stimulation improved gait performance by normalizing stride variability, walking speed, stride length, and cadence through enhanced sensory feedback [41]. In another study, the authors stated that the difference in touch thresholds they found between PD patients with and without a history of falling might be an association between reduced peripheral sensation and increased postural instability in the fallers group [282]. In summary, since the stimulation of the plantar foot enhances motor performance in PD patients, the link between reduced plantar sensation and motor symptoms in PD seems plausible. Hence, the reduced plantar cutaneous vibration perception found in our patient groups might be another factor contributing to motor symptoms in PD, such as postural instability or gait difficulties.

The causes of sensory symptoms in PD are multifactorial; however, there is a strong association with widespread deposits of α-synuclein, a fundamental pathological protein, which is also a major component of malicious Lewy bodies in PD patients [16,57,58,59,60,61,62]. Based on neuroanatomical models, the progression of α-synuclein might already begin in the prodromal stage of the disease, initiating in the lower brainstem, autonomic nervous system, and olfactory bulb, advancing in a caudal to rostral direction affecting the diencephalon, basal forebrain, medial temporal lobe structures, and finally neocortical areas [16,65,66,83,283,284,285]. While the progression of sensory symptoms in PD might be related not only to the extension of α-synuclein in specific dopaminergic structures, it also can be present in neurons, presynaptic terminals, and glial cells in the autonomic nervous system, the retina, the central and peripheral nervous systems, and therefore in epidermal nerves of the skin [57,63,64,65,67,68]. Studies have reported that sensory deficits are related to cutaneous denervation in PD, predominantly by α-synuclein. Investigating skin and cutaneous nerves from the abdominal wall in PD patients, Ikemura et al. found extensive Lewy body accumulation in up to 70% of the investigated cases with Lewy stages II and III, which corresponds to preclinical and early stages of PD [63]. Using skin biopsy to assess peripheral denervation, Nolano et al. found a lower density of intrapapillary myelinated endings of the glabrous skin in PD patients compared to healthy subjects. In both the glabrous and the hairy skin, the authors also observed axonal swelling and myelin abnormalities, such as paranodal and distal demyelination, profile segmentation and occasional internodal demyelination of the epidermal nerve fibers. This also includes myelinated axons of the fiber type Aβ which are responsible for conducting afferent tactile information from mechanoreceptors, such as the Meissner corpuscles, to the central nervous system [286]. More specifically, the number of Meissner corpuscles that detect mechanical vibration stimuli was significantly reduced. Furthermore, Meissner corpuscles even presented a wide range of anomalies, which, according to the authors, suggests the coexistence of degenerative and regenerative processes. The loss of Meissner corpuscles also correlated with the disease severity of the patients. The authors concluded that peripheral deafferentation, including Meissner corpuscles in PD patients, could play a major role in the pathogenesis of sensory dysfunction and could account (at least partly) for the impairment in sensory function in PD [64]. Since we predominantly investigated the functionality of Meissner corpuscles, the findings from Nolano et al. might therefore be one reasonable explanation for why our PD patients also showed impaired plantar cutaneous vibration sensitivity. As those studies mainly show degeneration and deficits of the peripheral nervous system in PD patients, there is also evidence for defective central integration and processing of afferent information at a cerebral level in PD. Although the basal ganglia are considered well-established primarily motor-related structures, there is conjoining clinical and experimental evidence supporting basal ganglia as active “sensory analyzers” for higher-level central somatosensory processing [69,70,71,72,73]. This is plausible, since the basal ganglia are connected to the cortex and receive input from not only motor areas but also cortical somatosensory areas [69,70]. Particularly the STN, one of the main input structures of the basal ganglia, receives projections from multiple cortical, predominantly sensorimotor, areas, whereas its disease-related hyperactivity might cause the loss of functional specificity and ultimately alter somatosensory and sensorimotor integration processing of tactile afferent information [36,287,288,289]. Boecker et al. investigated altered activity of various brain structures, including regional cerebral blood flow and metabolism, using 3D positron emission tomography while applying vibration stimuli to the skin of the index finger. Their results showed that sensory-evoked brain activation in PD patients was reduced in subcortical (basal ganglia) and cortical (parietal and frontal) areas compared to that of healthy control subjects. More specifically, PD patients showed decreased activation of the contralateral sensorimotor and lateral premotor cortex, the contralateral secondary somatosensory cortex, the contralateral posterior cingulate, the bilateral prefrontal cortex (Brodmann area 10), and the contralateral basal ganglia. In contrast, there was a relative enhanced activation of ipsilateral sensory cortical areas, notably caudal primary and secondary somatosensory cortices and the insular cortex, in PD patients compared to healthy subjects. The authors interpreted their findings as an indication of either altered central focusing and gating of afferent sensory impulses or enhanced compensatory recruitment of associative sensory areas in the presence of patients’ basal ganglia dysfunction. Hence, with their findings, Boecker et al. showed that basal ganglia dysfunction in PD is characterized by abnormal sensory processing, even for tasks devoid of any motor component [73]. In this context, other studies have also suggested that altered tactile perception, including impaired shape discrimination and tactile acuity, reduced roughness detection at the fingertips, altered two-point tactile discrimination thresholds and abnormal weight perception thresholds, are the result of defective central processing attributed to the diseased basal ganglia in PD [38,75,76,77,78,79,80,81,82]. It is assumed that the so-called “neural noise” in the somatosensory loops of the basal ganglia may also contribute to the increase in tactile detection thresholds [84,85,86,90]. Disease-related changes in the receptive fields for tactile inputs to the basal ganglia may introduce “noise” into sensory perception, resulting in increased thresholds and reduced discriminative capacities for different sensory modalities [84,85]. This might be emphasized by excessive pathological synchronous neural activity in the beta frequency band (8–35 Hz) throughout the cortico-basal ganglia network in PD patients. Accordingly, cortical oscillations in the beta-range “contaminate” the oscillatory activity of the basal ganglia and prevent their desynchronization, which is essential for movement control, but can possibly also play a role in sensory processing [86,290,291]. Besides impaired basal ganglia functionality, the dysfunction of extranigral pathways, including the brainstem nuclei, diencephalic and cortical areas, as well as extra-encephalic structures, such as the spinal cord and the autonomic enteric plexus, might be associated with sensory deficits in PD [61,90,292]. In summary, impaired tactile cutaneous perception in PD might be driven by denervation of peripheral epidermal nerve fibers and mechanoreceptors, as well as by defective central integration and processing of afferent information at a cerebral level. Therefore, those pathophysiological mechanisms might help to explain the increased plantar cutaneous vibration thresholds found in our patient groups. 

As the pathophysiological mechanisms mentioned above can develop inconsistently and therefore dominate either the left or the right cerebral hemispheres and body sides, this can cause laterality of symptoms. Since laterality is common for motor symptoms, such as tremor, it might also apply to sensory symptoms. This might at least be partly true for our findings, since we found an effect of a disease-dominant side in PD-MED. Patients with a disease-dominant left side showed higher vibration perception thresholds of the left foot compared to the right foot. This is supported by other studies, which also report laterality of sensory symptoms [36,64,83,293]. Nevertheless, we did not find this effect for patients in group PD-MED with a disease-dominant right side or for patients in group PD-MED–DBS, which is consistent with various other studies [34,37,81,87,89,95].

When analyzing whether anti-parkinsonian medication alone or medication in conjunction with DBS results in differences for tactile somatosensory functionality of the plantar skin in PD patients, we generally found higher vibration perception thresholds for group PD-MED compared to group PD-MED–DBS. Although the comparison between PD-MED and PD-MED–DBS had no statistical significance, our results showed a strong trend towards more impaired tactile perception for patients treated with anti-parkinsonian medication alone. This trend is also supported by the fact that the vibration thresholds only differed significantly between PD-MED and HS, while there was no difference between PD-MED–DBS and HS. Although the effect of anti-parkinsonian medication on sensory deficits, including noxious and innoxious tactile thresholds, and thermal perception in PD is controversial, reports about general insufficiency seem to dominate the literature [19,48,80,83,87,89,294]. For example, investigating plantar vibration perception thresholds in PD patients on and off medication, McKeown et al. found no acute effects of ceasing levodopa intake on plantar sensitivity [48]. Doty et al. also reported that plantar point pressure sensitivity thresholds were not affected by levodopa [89]. Moreover, Gierthmühlen et al. reported that levodopa did not influence detection thresholds or pain sensitivity [37]. Investigating pain perception as a sensory symptom in PD patients, insufficiency of medical treatment was also reported in another study with a large number of patients with early to moderate PD. In this epidemiological study, approx. 80% of PD patients reported no difference in pain between the on- and off-medication states [294]. It has even been reported that dopaminergic medication can worsen sensory symptoms in PD, such as proprioception, which might be related to medication-induced side effects due to heavy medication loads [19,89,265,295,296]. Although the processing of different sensory modalities, including proprioception, and noxious and innoxious tactile and thermal perception might not be the same, those studies show rather subtle effects of anti-parkinsonian medication on sensory symptoms and point towards little involvement of dopaminergic systems. Nevertheless, the contribution of dopaminergic systems to sensory symptoms in PD is still unclear. 

On the other hand, several studies, including this current study, have shown that DBS of the STN is more promising than anti-parkinsonian medication alone for treating sensory symptoms in PD [19,32,33,34,36,37,83,90,91,92,95,297,298]. For example, Cury et al. stated that DBS has a clear effect on sensory thresholds and changes sensory abnormalities towards normal levels in PD patients [83]. Aman et al., who investigated haptic discrimination thresholds of the hand, also reported enhancements of more than 20% with DBS compared to cases without DBS [36]. The authors concluded that improved haptic precision might indicate improved somatosensory functionality by STN-DBS. Their results support the findings from Maschke et al., who also showed a 20% decrease in position sense threshold as a result of DBS [36,298]. In a more recent study, Sabourin et al. investigated specific settings of directional DBS electrodes on sensory symptoms using a quantitative sensory testing battery, including thermal, pressure, and vibration perception. Although the effects were subtle, their results demonstrated that DBS modulates thermal and mechanical cutaneous sensitivity. DBS pulse width modulated mechanical sensitivity, whereas the DBS total electrical energy modulated thermal sensitivity when using certain directional contacts of the electrodes [32]. Altering the stimulation frequency of DBS, Belasen et al. also analyzed its effects on sensory modalities, including cutaneous pressure and vibration perception. The authors reported that lower DBS frequencies resulted in changed detection thresholds for mechanical pressure and vibration to a greater extent than higher frequencies [33]. In another study, Cury et al. reported lower thermal and mechanical detection thresholds post DBS surgery compared to those detected pre surgery. According to the authors, their data confirmed the existence of sensory abnormalities in PD and suggested that DBS mainly influences detection thresholds rather than painful sensations. In particular, DBS had a significant effect on mechanical and thermal detection thresholds, which were modified toward normal values after DBS surgery. Accordingly, DBS modulated both large and small fiber-dependent sensory input [90]. In contrast, the results of Ciampi de Andrade et al. showed that the detection of large fiber-mediated sensations, including vibration sensations at 100 Hz, did not change in PD patients between on-stim and off-stim conditions. However, PD patients had lower sensitivity to mechanical and thermal pain in the on-stim condition [34]. Dogru Huzmeli et al. also reported reduced thresholds of cutaneous two-point discrimination in PD patients after DBS, suggesting improved somatosensory processing [92]. Using questionnaires such as the non-motor symptom scale, several other studies also found STN-DBS to improve sensory symptoms in PD patients [299,300,301]. Those study findings support our results, showing that DBS is more efficient in treating sensory symptoms and normalizing tactile cutaneous perception thresholds compared to anti-parkinsonian medication alone. This becomes even more interesting when we consider that our study group that received DBS was affected for twice as long as the group that received medication alone. 

Since STN-DBS affects, first and foremost, the basal ganglia, changes in sensory perception are mainly associated with the modulation of somatosensory information at a cerebral level, while they probably have less effect on the peripheral nervous system per se [74,92,302,303,304,305,306]. The physiological mechanisms by which STN-DBS improves tactile cutaneous perception in PD patients remains unclear, but several hypotheses have been proposed. As STN-DBS acts on fibers and cells in close proximity to the implanted electrodes, an effect on specific somatosensory structures and pathways might be plausible, especially as the nearby thalamus plays a crucial role in processing sensory information [6,32,151,154,303]. In this context, it has been demonstrated that STN-DBS might modulate neural activity in the thalamus and other several cortical areas which are involved in processing tactile information [302,303,307]. Since the posterior parietal region receives information from prefrontal regions, the sensory cortex and multiple thalamic relay nuclei, STN-DBS may activate not only the frontal but also the parietal cortex, which suggests a contribution of the STN to sensory function [302,304]. The STN also has projections to the primary and secondary somatosensory cortices, which are responsible for processing tactile information, so that STN stimulation might affect sensory perception [32,74,302,304,305,308]. Using functional magnetic resonance imaging technology, DiMarzio et al. reported that the activity, especially of the primary somatosensory cortex, might be a promising indicator of whether sensory symptoms in PD patients respond to STN-DBS [308]. Another study has also mentioned that STN-DBS may alter the activity of the secondary somatosensory cortex, but this still has to be proven [74]. Three-dimensional positron emission tomography has also shown that DBS significantly increases the regional cerebral metabolic rate of glucose consumption in the frontal cortex, temporal cortex, parietal cortex, midbrain and basal ganglia, which may be associated with improved sensation [304,305]. Improved tactile perception through STN-DBS might therefore be the result of normalized inhibition–excitation communication of a comprehensive neuronal network, including numerous dopaminergic and nondopaminergic structures that are responsible for sensory processing [34,158,159,209,309]. Hence, DBS normalizes the disease-related hyperactivity of the STN, and consequently modulates the activation of the somatosensory cortex and enhances sensorimotor integration and processing of tactile afferent information [36,74,95,287,288,289,310]. It has been speculated that the high-frequency DBS signal overwrites the pathological activity of the STN, leading to dysfunction within the basal ganglia-thalamo-cortical connections [90,306,311]. Given the connectivity between the subthalamus, pallidus, and thalamus and the ascending projections into the somatosensory cortices, DBS-induced regulation of neuronal firing bursts that improve somatosensory processing seems plausible [36,90,265,266,311]. Thus, this mechanism is believed to restore the ability of thalamo-cortical relay cells to respond to depolarizing inputs involved in sensorimotor integration [312,313]. Moreover, it might reduce nigrostriatal “noise” and enhance the signal-to-noise ratio for a better signal discrimination, which is needed for tactile perception [36,90,311,314,315]. 

Since we found rather subtle effects of STN-DBS on tactile cutaneous perception thresholds compared to various other studies, we raised questions about the reasons for a lack of DBS efficacy. Hence, primarily methodological factors such as different surgical procedures and DBS settings must be considered when interpreting our results. In this context, Pötter-Nerger and Volkmann discussed the importance of distinguishing between a “primary” failure, attributed to suboptimal DBS settings, and a “secondary” failure, attributed to the diminishing benefits of stimulation due to disease progression [171]. Some of our STN-DBS patients were tested before their neurological consultation, while others were tested after. Consequently, those tested before the consultation did not benefit from potentially optimized DBS settings, introducing a potential bias toward lower efficacy in our results [180]. This could have been a potential issue in our study, considering that our STN-DBS patients had been suffering from the disease for more than twice as long as the patient group which received medication only. Furthermore, group PD-MED–DBS had relatively long STN stimulation intervals of approx. 3 years post surgery [98,99,100,171,179,180]. Further possible explanations include different study group compositions and the individual disease severity. Due to patients having high inter-individual symptom characteristics, a higher sample size might have been beneficial in detecting more robust group differences. Furthermore, conducting a longitudinal interventional study design to analyze tactile cutaneous perception before and after DBS surgery, with and without additional medication, instead of the cross-sectional design, could have provided a more accurate investigation of each therapy independently.

## 5. Conclusions

This study investigated axial motor symptoms and somatosensory functionality in PD patients and analyzed whether anti-parkinsonian medication in conjunction with STN-DBS shows different effects compared to medication alone. Healthy subjects were included as a reference. With respect to axial motor symptoms, we hypothesized that patients’ motor performance is worse compared to those of healthy subjects. Moreover, we assumed that medication in conjunction with additional STN-DBS is more advantageous in normalizing patients’ impaired motor performance compared to treatment with medication alone. 

Regardless of the therapy, PD patients in both study groups showed higher postural sway in the AP and ML directions during quiescent bipedal stance compared to the healthy subject group HS. No differences could be found between treatment conditions. The functional limits of stability test revealed smaller base of support ranges, especially characterized by restricted limits of stability in the posterior direction for both patient groups, PD-MED and PD-MED–DBS, compared to the healthy subject group, HS. Moreover, patients in both groups needed longer times and moved slower during the test. Nevertheless, PD patients who received medication in conjunction with STN-DBS showed significantly larger limits of stability in the anterior direction compared to patients who received anti-parkinsonian medication alone. Patient gait was mainly characterized by slowness due to shorter strides and fewer steps per minute, which also reflected longer strides and double support times compared to HS. PD patients also showed broader stride widths compared to HS. Only minor differences pointing towards better gait performance were found for PD-MED–DBS compared to PD-MED. To summarize those results, we found that PD patients suffer from impaired postural stability during quasi-static and dynamic balance situations, as well as from impaired gait performance compared to the healthy subject group, which therefore confirms our hypothesis. Anti-parkinsonian medication in combination with STN-DBS tended to be superior for improving patients’ overall motor performance compared to medication alone. This was especially true for the limits of stability and gait. Nevertheless, as there were no significant improvements in the majority of the investigated motor parameters, we reject our hypothesis that anti-parkinsonian medication in combination with STN-DBS is superior for treating axial motor symptoms compared to medication alone. The differences between our results and those of studies demonstrating clear DBS-induced positive effects on motor performance can mainly be attributed to methodological factors, as described earlier. Therefore, our results suggest that testing patients’ postural and gait performance should be challenging and quantify various different aspects of motor performance in order to evaluate patients’ axial motor symptoms more comprehensively. As prominent clinical markers of patients’ motor disability, the limits of stability in the anterior and posterior directions and gait speed should be observed more carefully. Further studies examining the impact of DBS and anti-parkinsonian medication on axial motor symptoms should involve varying combinations of anti-parkinsonian medication and varying stimulation parameters. Besides STN-DBS, the effects of stimulating other target areas, such as the globus pallidus interna and the pedunculopontine nucleus, on motor performance should also be investigated. We also recommend additional investigations, including EMG analysis of various muscles controlling movements of the ankle, knee, and the hip joints, as well as 3D motion analysis for analyzing full body motion.

The second objective of this study was to investigate somatosensory functionality by analyzing plantar cutaneous VPTs within the same study groups. Based on the pathophysiological mechanisms of PD and previous study findings, we hypothesized that patients’ plantar cutaneous vibration perception is impaired compared to healthy subjects. Since our results showed that the plantar cutaneous vibration perception of PD patients who received only anti-parkinsonian medication was significantly higher compared to that of the healthy subject group, we can confirm our hypothesis. This suggests that the pathophysiological mechanisms of PD affect plantar tactile cutaneous perception. Moreover, our hypothesis is supported by the results of various other studies that found impaired tactile cutaneous perception in PD patients by analyzing numerous body regions, including the torso, arms, hands, and fingers, and the feet. Consistent with the argumentation above that plantar mechanoreceptor input contributes to motor control and the study findings that textured insoles improve motor performance in PD patients, our results also suggest that impaired plantar cutaneous vibration perception might therefore contribute to axial motor symptoms in PD patients. Hence, our results may be helpful for developing and implementing plantar tactile cutaneous perception-enhancing therapy strategies. Moreover, they can be used to design and optimize low-cost therapy devices, such as textured insoles, to stimulate plantar cutaneous mechanoreceptors and consequently enhance motor performance in PD patients. As this is the first study investigating the effects of STN-DBS on plantar tactile cutaneous perception in PD patients, we furthermore hypothesized that anti-parkinsonian medication in combination with STN-DBS might show superior effects on normalizing patients’ impaired plantar cutaneous vibration perception compared to medication alone. Therefore, based on our results that STN-DBS improves plantar cutaneous vibration perception and on other study findings that also showed that STN-DBS improves tactile cutaneous perception, we confirm our hypothesis. Our results suggest that PD patients’ impaired plantar tactile cutaneous perception improves through STN-DBS, presumably by normalizing the integration and processing of afferent input on a higher-order cerebral level. Further studies examining the impact of DBS and anti-parkinsonian medication on plantar tactile cutaneous perception should involve varying combinations of anti-parkinsonian medication and varying stimulation parameters. Besides STN-DBS, the effects of stimulating other target areas, such as the globus pallidus interna and the pedunculopontine nucleus, on tactile cutaneous perception should also be investigated. Moreover, subsequent studies should focus on examining the functionality of various cutaneous mechanoreceptors that are sensitive to vibration stimuli at different frequencies, and to noxious and innoxious pressure and temperature stimuli. 

## Figures and Tables

**Figure 1 brainsci-13-01681-f001:**
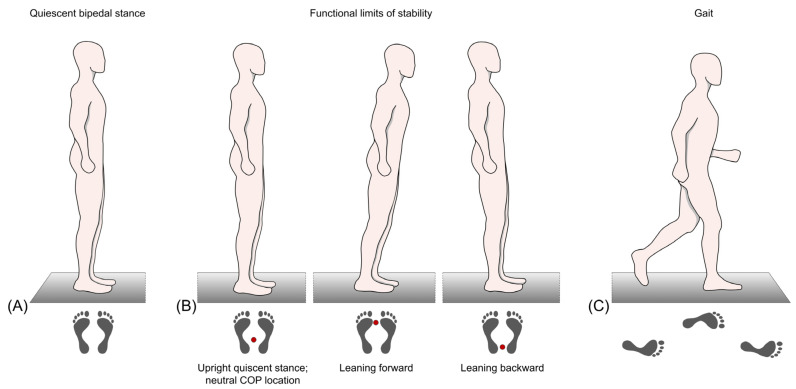
Testing subject motor performance on the pressure distribution platform: (**A**) Quiescent bipedal stance; (**B**) Functional limits of stability—the red dot indicates the location of the center of pressure (COP); (**C**) Gait.

**Figure 2 brainsci-13-01681-f002:**
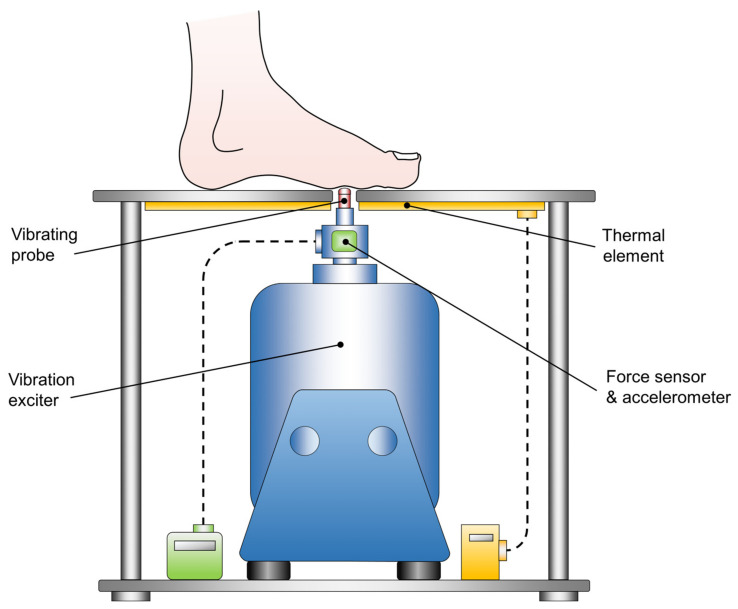
Examination of the plantar cutaneous vibration perception threshold (VPT) at the first metatarsal head.

**Figure 3 brainsci-13-01681-f003:**
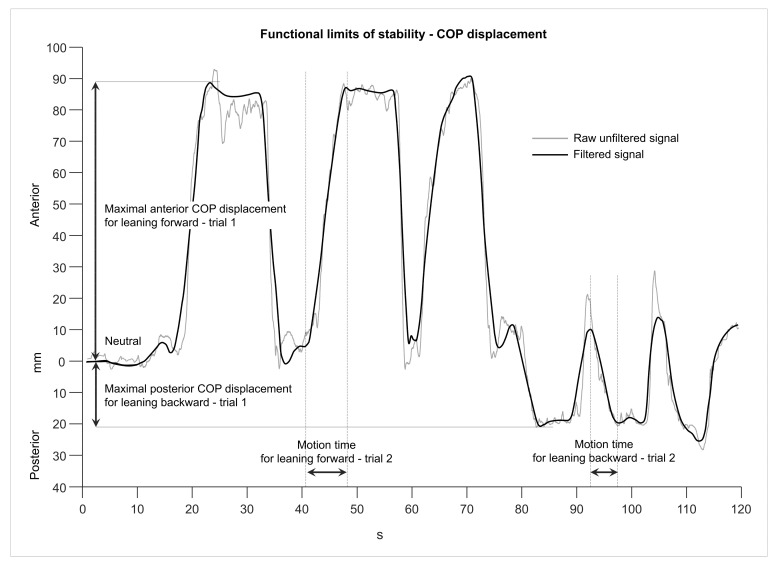
Visualization of an actual COP displacement signal from a randomly selected patient in study group PD-MED–DBS while performing the functional limits of a stability test (raw unfiltered signal: grey line; filtered signal: black line). The figure indicates the detection of the maximal COP displacement and the motion time in anterior and posterior directions.

**Figure 4 brainsci-13-01681-f004:**
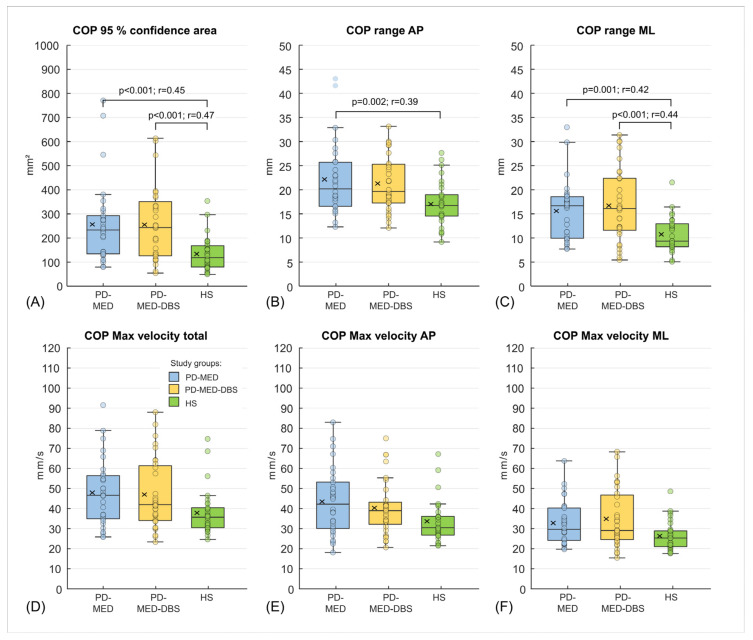
Group comparisons for COP parameters: (**A**) COP 95% confidence area; (**B**) COP range AP; (**C**) COP range ML; (**D**) COP max velocity total; (**E**) COP max velocity AP; (**F**) COP max velocity ML. The cross within each box marks the mean value. Statistically significant differences between groups (*p* < 0.0028) are indicated, as well as effect sizes, r.

**Figure 5 brainsci-13-01681-f005:**
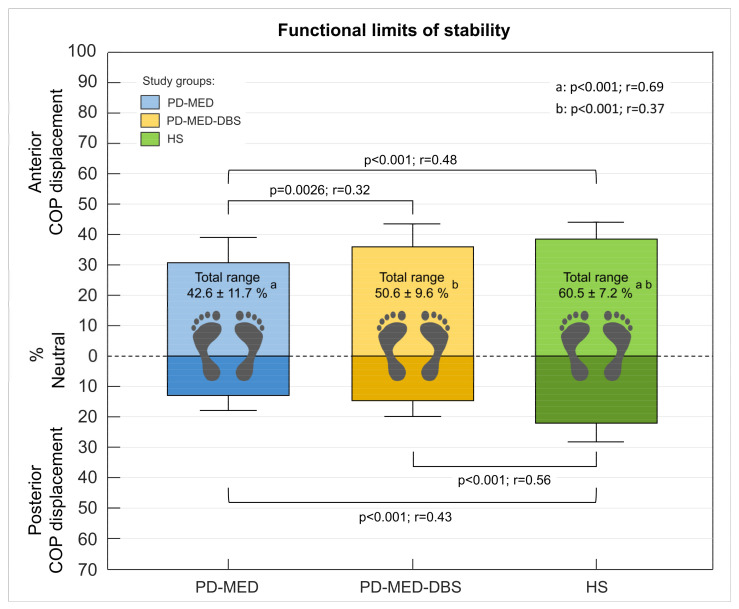
Group comparisons for the maximal COP displacements in the anterior and posterior directions and the total COP range during the functional limits of stability test. Values are shown as percentage of the foot length (Mean + SD). Statistically significant differences (*p* < 0.0028) are shown, as well as effect sizes, r.

**Figure 6 brainsci-13-01681-f006:**
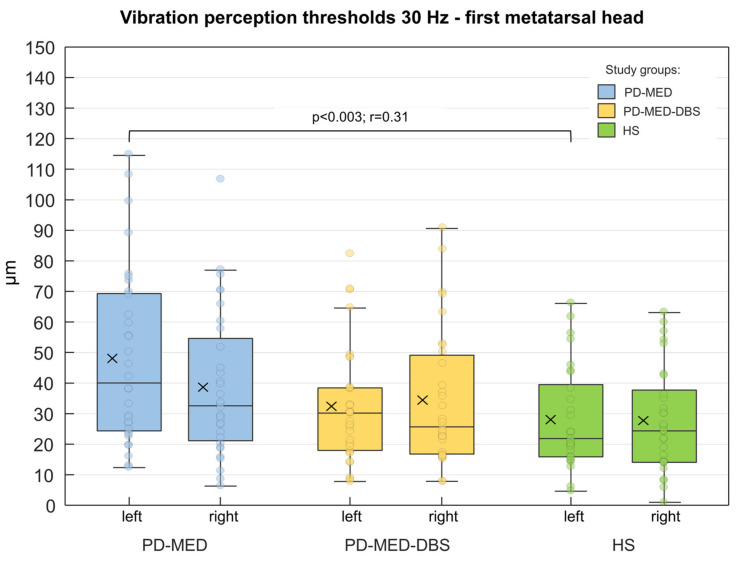
Group comparisons of the VPTs from the left and right first metatarsal head of the plantar foot. Displayed are the untransformed absolute VPTs. The cross within each box marks the mean value. Statistically significant differences (*p* < 0.0083) are shown as well as effect sizes, r.

**Table 1 brainsci-13-01681-t001:** Demographic and clinical data of the investigated study groups (mean ± SD); statistically significant differences for PD-MED vs. PD-MED–DBS vs. HS, *p* < 0.0167 and PD-MED vs. PD-MED–DBS, *p* < 0.05 are marked with a and b; r-values indicate the effect sizes of statistically significant group differences.

		PD-MED	PD-MED–DBS	HS	*p*-Values; r
Demographic data:	*n*/sex	37/♂ 29/♀ 8	39/♂ 24/♀ 15	30/♂ 19/♀ 11	–
	age (years)	68.5 ± 7.5	66.2 ± 6.5 ^b^	70.6 ± 5.7 ^b^	^b^ 0.023; 0.13
	height (cm)	169.6 ± 13.6	168.7 ± 8.1	171.3 ± 8.9	–
Clinical data:	self-rated balance confidence (0–100) (%)	60.0 ± 20.2 ^a^	59.8 ± 20.1 ^b^	80.7 ± 9.4 ^a b^	^a^ <0.001; 0.50 ^b^ <0.001; 0.43
	self-rated gait confidence (0–100) (%)	64.0 ± 22.5 ^a^	63.3 ± 18.3 ^b^	89.1 ± 8.4 ^a b^	^a^ <0.001; 0.55 ^b^ <0.001; 0.54
	MMSE (0–30)	28.5 ± 1.7	28.4 ± 1.6	–	–
	UPDRS III (0–108)	16.9 ± 7.9	15.7 ± 6.6	–	–
	UPDRS total (0–199)	27.8 ± 13.2	29.2 ± 10.7	–	–
	Hoehn and Yahr (0–5)	2.1 ± 0.5	2.1 ± 0.4	–	–
	disease duration since diagnosis (months)	85.1 ± 65.7	189.2 ± 77.6	–	–
	disease-dominant body side	left: 17; right: 20	left: 15; right: 24	–	–
	time btw last neurological examination and testing (months)	5.7 ± 15.2 ^a^	3.5 ± 3.9 ^a^	–	^a^ < 0.001; 0.66
	DBS duration since surgery (months)		33.1 ± 25.7	–	–
	self-rated satisfaction with DBS (%)		80.1 ± 20.6	–	–

**Table 2 brainsci-13-01681-t002:** Group comparisons for the temporal and spatial–temporal COP parameters of the functional limits of stability test. Statistically significant differences (*p* < 0.0028) are shown (a, b), as well as effect sizes, r.

	PD-MED	PD-MED-DBS	HS	*p*-Values; r
COP motion time anterior (s)	7.1 ± 2.6 ^a^	5.3 ± 2.1 ^b^	3.0 ± 1.1 ^a b^	^a^ <0.001; 0.68; ^b^ 0.002; 0.43
COP motion time posterior (s)	6.3 ± 2.5 ^a^	4.7 ± 1.7	3.2 ± 1.4 ^a^	^a^ <0.001; 0.56
COP mean motion velocity anterior (mm/s)	12.9 ± 7.9 ^a^	19.9 ± 10.9 ^b^	37.3 ± 14.0 ^a b^	^a^ <0.001; 0.71; ^b^ 0.002; 0.43
COP mean motion velocity posterior (mm/s)	6.2 ± 3.7 ^a^	8.3 ± 3.2 ^b^	21.4 ± 11.4 ^a b^	^a^ <0.001; 0.72; ^b^ <0.001; 0.49

**Table 3 brainsci-13-01681-t003:** Group comparisons of the gait parameters. Statistically significant differences (*p* < 0.0023) are shown (a, b), as well as effect sizes, r.

	PD-MED	PD-MED-DBS	HS	*p*-Values; r
Stride length (cm)	91.1 ± 18.0 ^a^	94.3 ± 12.8 ^b^	118.3 ± 16.1 ^a b^	^a^ <0.001; 0.56; ^b^ <0.001; 0.52
Stride width (cm)	10.7 ± 3.1	11.1 ± 3.0 ^b^	8.3 ± 1.8 ^b^	– ^b^ <0.001; 0.29
Stride time (s)	1.2 ± 0.2 ^a^	1.2 ± 0.1	1.1 ± 0.1 ^a^	^a^ <0.001; 0.39; –
Gait velocity (m/s)	0.8 ± 0.2 ^a^	0.8 ± 0.2 ^b^	1.1 ± 0.2 ^a b^	^a^ <0.001; 0.60; ^b^ <0.001; 0.49
Cadence (steps/minute)	49.2 ± 6.0 ^a^	52.4 ± 5.3	55.4 ± 4.6 ^a^	^a^ <0.001; 0.39; –
Double support time (%)	35.3 ± 7.1 ^a^	32.7 ± 4.6 ^b^	28.4 ± 3.7 ^a b^	^a^ <0.001; 0.47; ^b^ <0.001; 0.35
COP length mean left right (%)	84.5 ± 6.4 ^a^	85.8 ± 7.3	89.9 ± 6.2 ^a^	^a^ 0.002; 0.31; –

## Data Availability

The dataset used and analyzed in this study is available from the corresponding author upon reasonable request. The data are not publicly available due to privacy and ethical restrictions.

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
