# Peer review of "Does Impaired Plantar Cutaneous Vibration Perception Contribute to Axial Motor Symptoms in Parkinson’s Disease? Effects of Medication and Subthalamic Nucleus Deep Brain Stimulation"

_brainsci, 2023, doi:10.3390/brainsci13121681_

Round 1

Reviewer 1 Report

Comments and Suggestions for Authors

Introduction and discussion are too lengthy and contain much speculation and tangential concepts.  This paper should be divided into two; a review of the literature on the role of sensory input in motor PD symptoms, and a much more focused, concise report of the research itself.  Here, the research is almost lost in this 43 page manuscript....

There is no distinction made here between tremor dominant and gait/balance dominant forms of PD, as the authors point out in lines 433-435.  This is a very important distinction that one would expect to have a bearing on sensory input, and the subclassification of the subjects would be very helpful

There is almost certainly bias in selection of DBS patients that would skew the results when comparing to medication alone.  The significantly younger age of this group is a good demonstration of this problem, but it is not the only one.  There are many reasons for which a patient may or may not be a DBS candidate, some of which would be expected to affect posture and gait performance. 

Only STN DBS patients were studied, and not GPi, further reflecting a bias in selection, as most centers use these targets in different groups.  The STN target is actually well known to adversely affect balance in many patients

There is no attempt to examine either MED or MED+DBS patients in the “off” state which would be extremely informative.  In a study that wants to show the effect of treatment on a functional modality it is difficult to understand why this was not done.  It is normal and routine to test PD patients “off,” especially when DBS programming or medication adjustment is contemplated, so this would not have constituted a major ethical issue, but lack of this information greatly hampers interpretation of the study.

The improved function and vibration threshold for MED+DBS compared to MED alone is an association, especially in view of the lack of "off" testing and of selection bias in choosing only STN-DBS in the "on" state.  This does not demonstrate cause and effect.  This limitation should be prominently noted in the discussion.

Overall, I find this a very valuable contribution; but I would advise separating the extensive literature review and lengthy discussion into a separate paper, and reporting the research in a shorter, more concise format, that notes and justifies the rationale behind patient selection with regard to DBS candidacy, DBS target, and subcategory of PD.

Comments on the Quality of English Language

There are some oddities in the English usage and grammar, such as line 44 using the word "dramatic" instead of "significant" or describing Lewy bodies as "malicious," and there are a few sentences such as 49-51 that are not phrased correctly; in that instance the authors probably mean to say "While motor symptoms can be very prominent...PD can also be accompanied by..." Examples of this occur throughout the manuscript.  A simple style review or proof reading would help, but it is perfectly understandable as is

Reviewer 2 Report

Comments and Suggestions for Authors

I have some comments and suggestions regarding the article "Does Impaired Plantar Cutaneous Vibration Perception contribute to Axial Motor Symptoms in Parkinson’s Disease? Effects of Medication and Subthalamic Nucleus Deep Brain Stimulation."

Abstract: The abstract should not have subtitles such as "Objective." It would be more concise and reader-friendly to present the objectives without explicitly stating it as a subtitle.

Keywords: It is recommended to include simpler syntagms as keywords to improve the article's discoverability in online libraries.

Introduction: The introduction appears to be quite long and overly focused on generalities. It would be beneficial to streamline the introduction by providing more specific background information related to the study and its research question.

Results: The first table and Figure 3 should be moved to the Results chapter for better organization and clarity.

Discussion: The discussion section seems to be excessively lengthy and may divert the reader's attention with an overload of information from other articles. It is advisable for the authors to primarily highlight and compare the similarities and differences between their findings and those of relevant literature, rather than extensively discussing multiple sources. Adding some tables will be helpful.

Conclusion: The conclusion appears to be too long. To enhance readability and conciseness, the authors should consider trimming down the section while still effectively summarizing the main findings and their implications.

Limitations: It would be valuable for the authors to include a dedicated section on the limitations of their study. This would provide transparency and improve the overall quality of the research.

Overall, this study is engaging and offers valuable insights. However, the aforementioned suggestions would help enhance the clarity, coherence, and focus of the article.
